# To Boost or to Reset: The Role of Lactoferrin in Energy Metabolism

**DOI:** 10.3390/ijms242115925

**Published:** 2023-11-03

**Authors:** Giusi Ianiro, Antonella Niro, Luigi Rosa, Piera Valenti, Giovanni Musci, Antimo Cutone

**Affiliations:** 1Department of Biosciences and Territory, University of Molise, 86090 Pesche, Italy; giusy.ianiro@unimol.it (G.I.); a.niro2@studenti.unimol.it (A.N.); giovanni.musci@unimol.it (G.M.); 2Department of Public Health and Infectious Diseases, University of Rome La Sapienza, 00185 Rome, Italy; luigi.rosa@uniroma1.it (L.R.); piera.valenti@uniroma1.it (P.V.)

**Keywords:** lactoferrin, metabolic syndrome, glucose metabolism, lipid metabolism, iron metabolism

## Abstract

Many pathological conditions, including obesity, diabetes, hypertension, heart disease, and cancer, are associated with abnormal metabolic states. The progressive loss of metabolic control is commonly characterized by insulin resistance, atherogenic dyslipidemia, inflammation, central obesity, and hypertension, a cluster of metabolic dysregulations usually referred to as the “metabolic syndrome”. Recently, nutraceuticals have gained attention for the generalized perception that natural substances may be synonymous with health and balance, thus becoming favorable candidates for the adjuvant treatment of metabolic dysregulations. Among nutraceutical proteins, lactoferrin (Lf), an iron-binding glycoprotein of the innate immune system, has been widely recognized for its multifaceted activities and high tolerance. As this review shows, Lf can exert a dual role in human metabolism, either boosting or resetting it under physiological and pathological conditions, respectively. Lf consumption is safe and is associated with several benefits for human health, including the promotion of oral and gastrointestinal homeostasis, control of glucose and lipid metabolism, reduction of systemic inflammation, and regulation of iron absorption and balance. Overall, Lf can be recommended as a promising natural, completely non-toxic adjuvant for application as a long-term prophylaxis in the therapy for metabolic disorders, such as insulin resistance/type II diabetes and the metabolic syndrome.

## 1. Lactoferrin: General Aspects

In recent years, lactoferrin (Lf), an iron glycoprotein of the innate immune system, has received a great deal of attention for being one of the most promising and powerful natural products that can defend the host against various pathological conditions, including infections, inflammatory disorders, and cancer. It has been referred to as a “miracle molecule” [1] and a “Great Wall host defense” [2], and, in the wake of the coronavirus disease 2019 (COVID-19) pandemic outbreak, many reviews have been published on the potential beneficial effects of Lf in human health and disease. Here, an initial overview of the role of Lf in human energy metabolism will be provided, taking into account how the physicochemical characteristics of the glycoprotein may strictly influence its biological activity in this context. Given the large amount of available literature on this topic, only a brief introduction on Lf structure and its general functions follows.

Lf was first isolated from bovine milk in 1939 [3] and later identified in human milk in 1960 [4]. Lf is constitutively synthesized by exocrine glands and neutrophils following induction. Its highest concentrations are found in colostrum and milk, with lower levels in secretory fluids [5]. Human Lf (hLf) and bovine Lf (bLf) consist of 691 and 689 amino acid residues, respectively. Both reveal two symmetrical lobes, namely the N-lobe and C-lobe, connected by a short α-helix. Each lobe is further divided into two sub-lobes or domains, known as N1, N2, C1, and C2. These domains form a cleft in which the ferric ion (Fe^3+^) is tightly bound in coordination with a (bi)carbonate anion. Each lobe binds the ferric ion with high affinity through highly conserved residues: two tyrosines, an aspartic acid, and a histidine [6]. Other metal ions, such as Al^3+^, Cu^2+^, Mg^2+^, Mn^3+^, Zn^2+^, and Ca^2+^, are chelated by Lf through the same cleft, albeit with a lower affinity than Fe^3+^ [7].

Iron binding induces large conformational changes that define the open (iron unsaturated, apo-Lf) or closed (iron saturated, holo-Lf) state. Native Lf (nat-Lf) has a low rate of iron saturation (~10–20%), while holo-Lf is highly saturated, with more than 95% iron [6]. The iron-binding state of Lf can influence some of its functions. Compared with the apo form, holo-Lf has greater stability and resistance to thermal denaturation and protease digestion [8]. Lf can scavenge free iron in fluids and in inflamed or infected sites, helping to suppress free-radical-mediated damage and reducing iron availability to pathogens and cancer cells [9]. In this regard, recent efforts have been made to investigate the differential biological activities among apo-, nat-, and holo-Lfs, demonstrating that Lf can exert differential functions depending on its iron saturation rate [10,11]. Apo-Lf appears to be more effective in the antioxidant response by virtue of its ability to sequester ferric ions (Fe^3+^), thus protecting against the damaging effects of oxidative stress, specifically by limiting the production of hydroxyl radicals and peroxidized lipids [12,13]. A recent study by Ianiro and colleagues reported that the iron saturation rate influences the antioxidant and neuroprotective role of Lf in astrocytoma cells expressing the human immunodeficiency virus type 1 (HIV-1) Tat protein [14]. Bullen et al. [15] showed that the antibacterial activity of Lf decreases as the level of iron saturation increases. Subsequent in vitro experiments have shown that apo-Lf can inhibit the growth of several iron-dependent pathogens, such as *Escherichia coli*, *Vibrio cholerae*, and *Pseudomonas aeruginosa*, and that the addition of iron to the culture medium abolishes the bacteriostatic function of Lf [16]. In contrast, several in vitro studies have shown that both the apo- and metal-saturated forms of bLf exhibit comparable antiviral activity, suggesting that the inhibition of viral infection by Lf does not depend solely on its iron-chelating property [17,18]. Regarding the antiproliferative effect, it has been reported that recombinant human apo-Lf showed a more potent growth inhibition effect on HT-29 cells than did recombinant human holo-Lf [19]. Iron-saturated Lf exhibits superior antitumor property compared with low-iron apo-Lf. The study by Zhang and colleagues reported that holo-Lf induced ferroptosis in triple-negative breast cancer (TNBC) tumors by increasing the total iron content and promoting ROS production [20]. Furthermore, holo-bLf was found to be much more efficient in inhibiting glioblastoma migration than its native counterpart, both at the cellular and molecular levels [11]. Overall, all this evidence suggests that apo-Lf is more suited to be applied in contexts of oxidative stress and/or inflammation, whereas the iron-saturated one is better suited to the cancer field. Thus, a rigorous evaluation of the iron content in Lf formulations before any application is strictly needed.

The biological function of Lf is also affected by glycosylation, the most common post-translational modification (PTM) that affects protein folding, immunogenicity, protein solubility, and resistance to proteolysis [21]. The glycosylation status can change depending on the species, the cell type expressing the glycoprotein, the amino acid sequence, physiological conditions, and the purification strategy. HLf contains two major glycosylation sites, Asn 138 and Asn 479, but different patterns of N-glycans. In the case of bovine milk, the main source of commercial bLf, the distribution of glycans varies depending on the breed of the cow, the lactation period, the animal’s diet, and the glycoprotein purification strategy [21]. Five N-linked glycosylation sites are present on bLf (Asn233, Asn281, Asn368, Asn476, and Asn545), but only four sites are invariably glycosylated (Asn233, Asn368, Asn476, and Asn545) [22], whereas Asn281 can selectively undergo glycosylation, resulting in a bLf with higher molecular mass (84 kDa versus 81 kDa) [23]. Few studies have highlighted the functional contribution of glycans to Lf biological functions. Sialic acid has been acknowledged for its antithetical role in bLf antiviral activity against enveloped and non-enveloped viruses. In fact, it has been shown that the anti-rotavirus activity of bLf increases upon the removal of sialic acid [18], whereas desialylated bLf is less effective against the influenza A virus [24], thus suggesting that this opposite effect might be related to the different features of the viral particles themselves.

The broad spectrum of functions exerted by Lf is closely related to its high cationic charge at physiological pH. Antiviral, antimicrobial, and antitumor activities have been related to this feature, as they can promote the targeting of Lf toward viral, microbial, and tumor cell particles. Indeed, Lf is able to compete with viral and microbial attachment to host cells by competitively binding to specific sites, including negatively charged cellular compounds such as glycosaminoglycans (GAGs; mainly heparan and chondroitin sulfates) and sialic acids [25,26,27,28].

The ability to bind to cell-surface molecules and the discovery of specific, ubiquitous receptors that Lf can activate make this glycoprotein highly bioavailable, although many physiological and metabolic aspects have yet to be characterized.

The bioavailability of Lf represents a unique biological challenge, because mammals have variable exposure to endogenous Lf during different life stages. During infancy, Lf is abundant through breastfeeding, but its availability decreases during development and aging.

In healthy individuals, Lf concentrations in the blood are relatively low, ranging from 0.02 to 1.52 µg/mL, whereas in cases of infection, inflammation, excessive iron intake, or tumor growth, Lf levels may increase [29]. In vitro studies on the multifunctional activities of Lf, such as its anti-inflammatory and antitumor properties, have shown that higher concentrations of Lf are required to be effective. Therefore, in vivo administration of exogenous Lf is necessary to overcome this limitation. Excluding the consumption of Lf through milk, the natural intake of exogenous Lf in the diet is generally limited and varies according to individual habits. Currently, bLf has become readily available on the commercial market and is used in various applications. It is commonly found as a supplement in infant formula and added to various foods to support gut flora, improve iron absorption, and immune system function. Numerous animal and human tests have demonstrated the safety and tolerability of Lf, even at high doses, causing it to be classified as generally recognized as safe (GRAS) by the Food and Drug Administration (FDA) and as a dietary supplement by the European Food Safety Authority.

Several routes of administration have been explored, with oral administration being the most practical and safe approach [30]. Considering that many bLf functions (such as the ability to bind iron) are highly dependent on the integrity of the protein structure, gastrointestinal digestion results in the loss of many of these properties. Since bLf receptors are located in the intestine [31,32], the efficacy of bLf oral administration depends on the passage through the stomach to reach the intestine without significant degradation. To improve Lf oral bioavailability, different methods such as microencapsulation, PEGylation, and absorption enhancers have been proposed [33].

The bioavailability of orally administered Lf varies with age, with children showing higher intestinal absorption rates than adults. Survival of Lf during gastrointestinal passaging is higher in neonates due to a leakier gut, which allows it to be safely transferred from the gut into the systemic circulation. To improve the bioavailability of bovine Lf, oral administration should occur before meals to avoid gastric degradation [34].

The initial absorption of Lf occurs through specific Lf receptors of the intestinal brush border. In vitro studies in human cell lines show that there can be dual transport mechanisms, with a major degradative pathway leading to the transcellular transport of degraded fragments and a minor pathway allowing the passage of immunoreactive Lf [35].

Lf passes from the stomach into the blood through epithelial cells by endocytosis [36,37], mainly via Peyer’s patches [38]. Once in the intestine, Lf can interact with its receptor and be internalized by enterocytes, then partially carried into the bloodstream and delivered to tissues. Thus, absorption occurs mainly through the lymphatic circulatory system rather than through portal circulation [39,40].

Lf can also be reabsorbed by bile [36,41] or transported to the central nervous system (CNS) via the cerebrospinal fluid [42,43] and across the blood–brain barrier [44]. After binding to the cognate receptor, Lf is internalized through clathrin-mediated endocytosis and translocated into the nucleus [45].

Blood Lf is cleared from the circulation by the liver, primarily by hepatocytes but also by Kupffer cells and the liver endothelium [41,46,47,48]. A role of the spleen and the kidneys in Lf clearance has also been demonstrated [49].

Lf performs its functions after interacting with Lf-specific receptors (LfRs) present in different tissues and cell types, including hepatocytes, kidney cells, lymphocytes, monocytes, and various regions of the brain [50].

Intelectin 1 (ITLN1) is a lectin located on the brush-border membrane of the small intestine that plays a role in the innate immune response and also acts as a high affinity receptor for Lf. It is a 120 kDa homotrimer glycoprotein in which three 40-kDa subunits are cross-linked by disulfide bonds [51]. ITLN1 has been found, other than in the intestinal brush border, in Paneth and goblet cells [52] as well as in the biliary epithelium [53]. Moreover, low-density lipoprotein (LDL)-receptor-related protein 1 (LRP1) can bind multiple targets including Lf, and it is principally expressed in hepatocytes, neurons, smooth muscle cells, fibroblasts, and cholangiocytes [53,54]. It is a 600 kDa type I transmembrane receptor composed of five subunits, which mediates the endocytosis of diverse ligands, including proteases, growth factors, extracellular matrix proteins, lipoproteins, and other membrane proteins [55]. In hepatocytes, it is involved in the endocytic uptake of lipoproteins containing triglycerides and cholesterol. LRP1 also takes part in several cellular processes, including cell migration, survival, motility, and differentiation, and it has been proven to play a role in different pathologies such as thrombosis, fibrinolysis, and atherosclerosis [56].

Other types of LfRs have been characterized, including the asialoglycoprotein receptor (ASGPR) in liver, CD14 in monocytes, and nucleolin in lymphocytes. ASGPR is exclusively found in the liver. It is a carbohydrate-specific type C lectin that functions as a calcium-dependent receptor and is mainly present in the plasma membrane of hepatocytes [57]. Lf binds with high affinity (K_d_ ca. 80 nM) and in a galactose-independent manner to the rat hepatic lectin 1 (RHL1), the major subunit of the ASGPR, near the carbohydrate-recognition domain (CRD) on ASGPs [58,59]. CD14 is a 55 kDa glycoprotein that exists both as a soluble protein found in the serum at concentrations of 2–6 µg/mL and as a GPI-anchored protein (mCD14) on the surface of monocytes–macrophages [50]. It has been shown that hLf interacts directly with soluble CD14 (sCD14) and protects animals from septic shock induced by lipopolysaccharides (LPS) [60]. Nucleolin is a prominent 105 kDa nucleolar protein involved in fundamental aspects of transcription regulation, cell proliferation and growth. Found in rapidly growing eukaryotic cells, it acts as a cell-surface receptor for various ligands, including matrix laminin-1, midkine, attachment factor J, apo-B, and apo-E lipoproteins [61,62,63,64,65]. Recent studies have shown that nucleolin acts as a shuttle between the cell surface and the nucleus [65,66,67], suggesting its role as a mediator for the extracellular regulation of nuclear events. Interestingly, in addition to ITLN1, surface nucleolin is involved in the ability of hLf to enter the nucleus [68,69,70].

The existence of multiple receptors arguably underpins the substantial and widespread effects that Lf can exert.

## 2. Lactoferrin and Human Metabolism

Metabolic pathways are tightly regulated [71], and many pathological conditions, including obesity, diabetes, hypertension, heart disease, and cancer, are associated with abnormal metabolic states [72,73]. The cluster of interrelated adverse metabolic markers of hyperglycemia, dyslipidemia, and hypertension, alongside central or abdominal obesity, is termed the “metabolic syndrome”, for which numerous strategies have been tested, including pharmacological treatments, physical exercise, and dietary regimens [74].

Nutraceuticals, assumed as a food supplement, besides serving as dietary nutrients, can act as cis- or trans-regulators of the human metabolome. Many of these functions occur through both the direct regulation of receptor-mediated cellular signaling, mainly in enterocytes and infiltrating immune cells, and the indirect boosting of mucosal microbiota, thus potentially ameliorating intestinal barrier integrity and homeostasis [75]. Other biological activities are carried out by peptide sequences encrypted inside a protein nutraceutical, which exert their actions when released, mainly enzymatically, during food processing, digestion, or microbial fermentation [76].

Whey proteins, as key constituents of milk-based products, have been assessed for their contribution to the potential beneficial effects of dairy consumption, as evidenced from studies which have demonstrated the efficacy of whey proteins in counteracting obesity-related pathologies. The ingestion of milk-derived proteins can affect plasma lipid levels; in particular, whey protein tends to lower plasma lipid levels [77] and to enhance weight and body-fat loss during energy restriction and limited hepatic fat accumulation [78]. Similarly, Zapata and colleagues found that whey protein and, even more so, the isolated components lactalbumin (La) and Lf, were able to induce sustained weight and fat loss and to ameliorate the energy expenditure in diet-induced obese rats, together with decreasing the levels of plasma leptin and insulin and improving glucose clearance, as well as decreasing hepatic lipidosis [79].

Among the whey proteins, Lf plays a central role in being naturally able to interact with host cell receptors and with microbial pathogen-associated molecular patterns (PAMPs), thus exerting a global multitargeted action. At variance with other whey proteins, Lf is also endogenously expressed by specialized host cells, hence representing a self-defense system which can be turned on/off depending on the metabolic circumstances. However, the tissue or plasma Lf concentrations required to exert a specific biological activity are hardly reached in vivo, therefore requiring additional administration from exogenous sources.

The first evidence for the association between endogenous circulating Lf and metabolic disorders was provided by two studies by Fernandez-Real and colleagues, who analyzed the concentration of circulating Lf in patients suffering from insulin resistance and altered glucose tolerance (AGT) [80,81]. In humans, both circulating Lf and Lf gene polymorphisms were shown to be associated with dyslipidemia and vascular reactivity in patients with AGT [80]. Moreover, the mean circulating Lf was significantly higher in healthy subjects than in AGT subjects, and it positively correlated with insulin sensitivity and negatively with age, fasting glucose, and glycated hemoglobin levels [81]. Interestingly, opposite results were obtained by Mayeur and colleagues, who reported a positive correlation between serum Lf and insulin resistance in a cohort of lean to moderately obese women [82]. Similarly, an increased baseline concentration of serum Lf, reflecting neutrophil priming caused by hyperglycemia, was described as a predictor of the long-term risk of fatal ischemic heart disease in newly diagnosed diabetes subjects [83]. Few studies have investigated the potential correlation between endogenous Lf changes in humans presenting with lipid metabolic disorders. A study on subjects with morbid obesity demonstrated a significant inverse association of circulating Lf with postprandial lipemia and oxidative stress after acute fat intake. Subjects showing the highest increase in serum Lf presented the lowest changes in free fatty acids (FFAs), high-density lipoprotein cholesterol (HDL-C), as well as in C-reactive protein and antioxidant enzymes such as catalase and glutathione reductase, suggesting an ameliorated response to fat load in those subjects with increased endogenous Lf [84]. On the other hand, no significant difference in serum Lf concentrations was found in metabolically healthy and unhealthy obese women, with only minor correlations with some anthropometric and metabolic parameters [85].

Globally, these contradictory results can be due to differences in sex, age, and related pathologies, or linked to the applied analysis and statistics. Moreover, most of these studies did not properly consider the contribution of exogenous sources from an individual diet regimen to Lf availability and levels in the gastrointestinal tract and blood circulation, thus posing a serious limitation to comparative analysis. Interestingly, Lf and Lf receptor gene variants have been associated with the prevalence of disorders in glucose and lipid metabolism. Two Lf gene polymorphisms (LTF rs1126477 and rs1126478) were reported to be associated with HDL-C and triglyceride (TG) levels in subjects with AGT [80], whereas significant differences in low-density lipoprotein cholesterol (LDL-C) levels between LTF rs1126477 gene variants were found in metabolically healthy obese (MHO) subjects [86]. A gene polymorphism in LRP1 (rs4759277) was also associated with fasting insulin levels and homeostasis model assessment of insulin resistance in patients with metabolic syndrome [87], whereas significant differences in waist circumference and HDL-C levels among rs4759277 gene variants were reported in MHO patients [86]. Taken together, these data do not support an exhaustive conclusion, and it is difficult to form a clear link between LTF variants and metabolic disorders. Hence, more approaches are still required to better clarify if an actual correlation between fluctuations in endogenous Lf levels and metabolic disorders exists and what the physiological significance of this could be.

More robust evidence for the effects of the exogenous administration of Lf on energy metabolism was shown in several in vitro animal models and a few clinical studies. Among the multiple intricate pathways constituting human metabolism, glucose and lipid metabolisms were the most investigated.

### 2.1. Lactoferrin and Glucose Metabolism

Human tissues depend on glucose as the major energy source, but excessive consumption can trigger acute and chronic metabolic disorders. Production and release of pancreatic hormones, mainly insulin and glucagon, ensures balanced blood levels of glucose.

In this scenario, Lf is intriguing as it can regulate glucose metabolism through both cis- and trans-acting mechanisms, which encompass the direct binding of circulating sugars or the upstream modulation of metabolic pathways. In early 2010, Mir and colleagues demonstrated the presence of a sugar-binding site in bLf, with a K_d_ for glucose in the range of 10^−4^–10^−5^ M [88]. Crystallographic studies showed that glucose and other sugars bind to bLf within an elongated shallow cleft on the surface of the C-lobe through several hydrogen bonds and van der Waals interactions [88]. The site has mixed hydrophilic and hydrophobic features, with polar residues (Thr, His, and Glu) in proximity to the two ends and non-polar residues (Pro and Gly) interspersed within the cleft. Of note, the authors showed that purified bLf C-lobe significantly decreased the level of free serum glucose in human blood in vitro. A deeper analysis of the effect of glucose binding to bLf on conformational changes and thermodynamic stability of the glycoprotein showed that glucose induces conformational changes in bLf like those observed in the typical heat-induced two-step denaturation process, with an initial increase in α-helix and β-sheet contents of 6% and 14%, respectively [89]. Interestingly, glucose impaired the proliferative effect of bLf toward MC 3T3-E1 cells in a dose-dependent manner, thus suggesting that free sugar can influence Lf biological activities [89]. To our knowledge, no sugar-binding properties have been shown for other members of the transferrin (Tf) family, suggesting that the mechanism selectively evolved in Lf, whose endogenous expression is firmly regulated by stress conditions. Overall, Lf’s ability to directly bind mono- and disaccharides could be exploited to reduce the presence, and thus the absorption, of free sugars in the gut during meals; however, more efforts are needed to clarify this hypothesis.

More reports have investigated the efficacy of exogenously administered Lf in alleviating disorders of glucose homeostasis. First, hLf was found to significantly boost the insulin-induced response, in terms of ^473Ser^Akt phosphorylation, in human hepatocarcinoma HepG2 and in non-differentiated and pre-differentiated 3T3-L1 fibroblast mouse cell lines under non-inflammatory and inflammatory conditions [90]. At a dose of 1 μM and higher, hLf showed insulinotropic activity, increasing Akt phosphorylation on serine 473 in conditions where insulin action was blunted, such as in pre-differentiated 3T3-L1 cells and in cells treated with pro-inflammatory media [90]. The potential of Lf in counteracting glucose disorders has been shown in two different studies by the group of Takeuchi, who investigated the effect of bLf administration in rats under physiological or hyperglycemia-mimicking conditions. Intraperitoneal administration of 100 mg/kg bLf to rats before an oral glucose-tolerance test (OGTT), following 60 min of restraint stress (RS) load, significantly lowered plasma glucose transition when compared with a vehicle-treated group. In parallel, bLf decreased plasma corticosterone, an RS-induced glucocorticoid known to promote glucose intolerance, while not affecting plasma levels of glucagon and insulin [91]. These findings suggest that the hypoglycemic activity of Lf could be linked to the improvement of insulin resistance, which can be attributed to the attenuated activation of the corticosterone axis. A second study was carried out to rule out the effects of the intraperitoneal administration of bLf in rats before intravenous glucose injection (intravenous glucose-tolerance test, IVGTT) vs. OGTT, discounting the factor of glucose absorption from the small intestine. Interestingly, bLf did not affect any glycemic parameters in IVGTT, while it reduced plasma glucose and maintained insulin levels in OGTT. These latter effects were associated with lower levels of the plasma glucose-dependent insulinotropic polypeptide (GIP) and to an increase in total plasma glucagon-like peptide-1 (GLP-1) in the bLf-treated group compared with saline-treated controls [92]. In addition, bLf administration was found to increase the absorption of glucose into the everted jejunum sac [92]. All these observations support the idea that bLf can exert its hypoglycemic function by sustaining insulin secretion via GLP-1 upregulation while, at the same time, leaving glucose absorption from the intestine unaffected. Within this framework, one study suggests that Lf can directly contribute to the enhancement of glucose transport into small-intestinal epithelial cells by sodium-dependent glucose transporter (SGLT) 1, by downregulating the Ca^2+^ and cAMP signaling pathways [93]. A more recent study on a murine model of streptozotocin-induced type 2 diabetes mellitus (T2DM) reported the efficacy of bLf administration in decreasing serum concentrations of glycated serum protein and fasting insulin, and increasing liver insulin sensitivity, mainly through the upregulated expression of the insulin receptor (IR), insulin receptor substrate (IRS)-1, GLUT-4, phosphatidyl inositol 3-kinase (PI3K) and Akt. Moreover, bLf reversed the abnormal pancreatic islets of diabetic mice by reducing oxidative stress and inflammation responses, as evidenced by the downregulation of tumor necrosis factor-α (TNF-α), interleukin (IL)-6, and IL-1β in the serum or liver [94]. These latter results are in line with the sole in vivo study on type 2 diabetic pediatric patients, where camel Lf (cLf) was tested for its potential antidiabetic effect [95]. Sixty young obese patients with type 2 diabetes treated with standard antidiabetic therapy were divided into two arms, one of which was orally administered with cLf capsules (250 mg/day, p.o) for 3 months. CLf treatment decreased diabetes-associated inflammation by inhibiting the Toll-like receptor (TLR)-4–NF-kB–SIRT-1 axis with a corresponding reduction in the serum pro-inflammatory cytokines IL-1β, IL-6, and TNF-α. Consequently, a significant increase in insulin expression and a concomitant decrease in serum glucose were found [95]. This was the first study to point out an association between the Lf-induced hypoglycemic effect and an increase in peroxisome proliferator-activated receptor (PPAR)γ expression, suggesting that the antidiabetic effects are due to different features of Lf, from its anti-inflammatory properties to its ability to boost insulin signaling and its insulin-sensitizing efficacy through the PPARγ-dependent cascade. This latter mechanism is involved in glucose homeostasis by both upregulating the liver glucokinase, the rate-limiting enzyme of glycolysis, and stimulating the hepatic uptake of glucose. Overall, these data indicate that Lf supplementation may play an important role in counteracting a glucose imbalance, whether related or unrelated to type 2 diabetes (Figure 1).

### 2.2. Lactoferrin and Lipid Metabolism

Lipids are the energy reserves of animals and perform various functions, from acting as chemical messengers to the maintenance of body temperature and key constituents of cell membranes [96,97]. To enter the circulation, dietary fats are first emulsified via bile acids (BA) and then absorbed by enterocytes, where they are resynthesized and packed into lipoprotein particles. During their journey through the vascular system, nascent chylomicrons lose two minor apoproteins (ApoA-I and ApoA-IV) that are replaced by ApoE and ApoC-II, which are crucial for further chylomicron processing. In fact, ApoC-II activates adipocyte lipoprotein lipase (LPL), which facilitates the digestion of the chylomicron TG into fatty acids (FA) and glycerol, while ApoE is recognized by the hepatocyte LDL receptor, the LDL receptor-related protein (LRP), and scavenger receptor B-1, which facilitate the endocytic uptake of the chylomicron remnants [98].

FA oxidation and de novo synthesis, as well as the expression of fatty acid transport proteins (FATPs), are closely regulated by various nuclear receptors, such as PPARα, PPARγ, and the bile acid receptor/farnesoid X receptor (FXR) [99].

Glucose and lipid metabolism are linked in many ways. Insulin affects the de novo synthesis of lipids at multiple levels, via the induction of lipogenic genes, the activation of sterol regulatory element binding protein 1c (SREBP-1c), and the Akt-regulated production of very-low-density lipoproteins (VLDLs) [99]. It is well known that diabetic patients often present with a typical dyslipidemia, characterized by elevated triglycerides, low HDL-C, and a predominance of small dense LDL particles. The existence of a link is further supported by the fact that insulin stimulates fatty acid synthase (FAS) expression via the PI3K pathway. At a transcriptional level, SREBP-1c and carbohydrate-responsive-element-binding protein (ChREBP), a glucose-dependent transcription factor, synergistically induce the expression of FAS and acetyl-CoA carboxylase (ACC) [98].

The potential of Lf in rebalancing lipid dysmetabolism has been investigated in several studies using in vitro and animal models, with few clinical approaches. From the encouraging evidence on the application of whey proteins in this field, bLf was first demonstrated in the early 2000s to reduce plasma and hepatic triacylglycerol and non-esterified fatty acids (NEFA), with a parallel increase in HDL-C levels, in mice fed with a standard diet [100]. Curiously, the same results were not recorded in mice fed with a high-fat diet (HFD), suggesting that Lf is only partially efficient. Similar results were obtained in a comparable study, where bLf intragastric administration was found to decrease lipid accumulation in the liver and mesenteric fat in mice [101]. On the other hand, in a model of rat fed with a high-cholesterol diet (HCD), intragastric administration of bLf significantly ameliorated homocysteine and leptin levels and decreased serum LDL-C and total cholesterol, while increasing serum HDL-C, as evidenced by the up-regulation of ApoA-I expression [102]. Similarly, the oral administration of enteric-coated bLf tablets significantly reduced total and LDL cholesterol levels in the serum, without interfering with HDL-C levels, in a high-fat and high-cholesterol diet model. Moreover, both LDLR and 3-hydroxy-3-methyl glutaryl CoA reductase (HMGCR) were significantly upregulated upon bLf treatment, and their transcriptional factor SREBP-2 also displayed a tendency toward upregulation [103]. These studies have reaffirmed, once again, how crucial the choice of the experimental model is to verify/deny the effectiveness of a compound and the associated biological activity. In C57Bl/6J mice, Lf supplementation enhanced weight loss, suppressed weight re-gain and protected against the development of fatty liver formation, but it also ameliorated glucose tolerance and adipocyte tissue inflammation, without interfering with energy intake [78]. Other animal studies have evidenced the protective role of oral bLf administration against the metabolic dysregulation associated with bacterial LPS [104] and intestinal flora dysbiosis [105,106]. In these latter papers, bLf was found to be able to rebalance both glucose and lipid metabolic disorders and to restore inflammatory parameters, thus highlighting the multifaceted mechanism of action of the glycoprotein [104,105,106]. Within the same framework, bLf was found to be an agonist of inulin, a prebiotic soluble fiber produced by many plants, in decreasing energy intake, body weight, fat and lean mass, along with the respiratory quotient, in diet-induced obese rats [107]. Of note, Lf was also found to inhibit dietary cholesterol absorption, while increasing its fecal excretion via interactions with bile acids in rats [108]. This effect was investigated in detail in C57BL/6J mice fed with high-fat/high-cholesterol diet containing cholate, where Lf was demonstrated to promote the expression of enzymes involved in both BA synthesis and conjugation, as well as to hinder hepatic cholesterol deposition through downregulation of the FXR-mediated enterohepatic axis [109]. Interestingly, in ApoE^−/−^ mice, a model for studying cholesterol-induced atherosclerosis, Lf supplementation was able to revert the effect of a high-fat and high-cholesterol diet by significantly decreasing serum and hepatic cholesterol levels and the average lesion area in the whole aorta while increasing fecal cholesterol contents. These results were associated with the Lf-mediated downregulation of hepatic HMG-CoA reductase and the upregulation of cholesterol 7-alpha hydroxylase, two rate-limiting enzymes involved in the synthesis of cholesterol and bile acids, respectively [109].

At a mechanistic level, Lf anti-adipogenic effects were investigated in several papers [81,110,111,112]. All these studies have demonstrated the ability of Lf to suppress the adipogenic differentiation of mouse pre-adipocyte MC3T3-G2/PA6 cells [110], mouse embryonic 3T3-L1 cells [90], and rat mesenteric fat-derived pre- and mature adipocytes [111,112]. Upon Lf treatment, in addition to the decreased levels of lipid droplets, CCAAT/enhancer-binding protein alpha (C/EBP-α) and PPARγ pathways were commonly found to be downregulated [90,110,111,112] along with other specific markers of adipogenesis, such as aP2 and adiponectin [110], perilipin and the cAMP axis [112], or acetyl-coenzyme A carboxylase alpha (ACC) and fatty acid synthase (FASN) via activation of the AMPK and Rb pathways [90]. Proteomic analysis in mature adipocytes from primary cultures of rat mesenteric adipocytes confirmed the direct involvement of cAMP and extracellular signal-regulated kinase (ERK) pathways and the downstream activation of CREB in the Lf-induced activation of HSL, a key enzyme catalyzing the rate-limiting step of lipolysis via LRP1 [113].

On the other hand, the proadipogenic effect of bLf was demonstrated in a study on human subcutaneous and visceral preadipocytes. HLf was found to activate proadipogenic genes, including PPARγ, ACC, FASN and adiponectin, and inhibit both AMPK/Rb signaling cascades and the pro-inflammatory cytokines interleukin-8 (IL-8), IL-6 and monocyte chemoattractant protein-1 (MCP1) [114]. The proadipogenic role of Lf was also confirmed by the same group in a human adipocyte model of LTF knockdown, where the downregulation of endogenous Lf was associated with decreased adipogenic, lipogenic, and insulin-signaling-related gene expression and a significant increase in the gene expression of inflammatory mediators [115].

These divergent results on the anti-/pro-adipogenic activity of Lf could be ascribed both to the source of Lf (bovine vs. human) and to the different cell models tested, as transformed cells could respond differently than primary cultures to external stimuli, including Lf [9]. In reprogrammed human brown adipocytes generated using PLAT-GP cells, bLf addition promoted energy expenditure and oxygen consumption by upregulating uncoupling protein 1 (UCP1) expression through the cAMP-PKA signaling pathway via the LRP1 receptor [116], thus reinforcing the proposal for the dietary application of Lf as a general booster of energy metabolism.

Nearly the same action mode and mechanistic processes were found to be exploited by Lf in animal models of dyslipidemia. In 2018, two studies reported the effects of orally administered Lf in high-fat diet-induced obese C57BL/6J mice [117,118]. Although both studies showed improvements, upon Lf treatment, in physical and serological parameters, such as visceral fat deposition, hepatic steatosis, lipid and glucose serum levels, along with a significant decrease in the expression of some lipogenic enzymes, such as SREBP-1, FAS. and ACC, differences in the activation of AMPK signaling were found. In line with in vitro models, Min and colleagues reported an increase in the p-AMPK/AMPK ratio following Lf treatment [118], but no significant difference in p-AMPK/AMPK and PPARα levels were found between HFD and HFD + Lf groups in the study by Xiong and colleagues [117]. In the latter study, apart from the significant downregulation of MCP-1, a chemokine involved in the progression of hepatic inflammation and fibrosis, Lf treatment was not found to be able to induce a significant decrease in mRNA levels of both IL-6 and TNF-α, two cytokines strictly interconnected with the AMPK cascade that was shown to be able to inhibit TNF-α-stimulated IKK/NF-κB and IL-6-induced JAK/STAT3 signaling pathways [119]. Altogether, these data could suggest that Lf may regulate hepatic lipid metabolism via pathways other than the AMPK signaling cascade; however, more efforts are required to better test and prove this hypothesis.

A recent metabolomic study was carried out to investigate the putative beneficial effects of early-life Lf administration in newborn piglets [120]. Pathway analysis of the metabolomic profiles showed that the Lf treatment affected lipid and amino acid metabolism in the liver, with a decrease in plasma urea nitrogen and increased levels of plasma albumin. In addition, Lf-treated piglets showed upregulated expression of β-oxidation-related gene (CPT1) and hepatic TCA and increased phosphorylation levels of mTOR and p70S6 kinase as well as increased levels of proteolytic enzymes and amino acid transporters in the jejunum. Further, Lf treatment decreased malondialdehyde levels and increased levels of hepatic antioxidant glutathione (GSH), glutathione peroxidase (GSH-Px), and glutamate cysteine ligase catalytic subunit (GCLC) [120], thus primarily evidencing positive effects of Lf consumption in the modulation of metabolic and defense pathways during early life.

Finally, a double-blind, placebo-controlled design study carried out in humans showed a potent anti-obesity effect of enteric-coated bLf tablets (300 mg/day) in decreasing visceral fat accumulation after an 8-week administration to Japanese men and women with abdominal obesity. Of importance, no adverse effects regarding blood lipid or biochemical parameters were recorded in the bLf-treated group [121], thus highlighting, once again, the global safety of this molecule and encouraging its application as a promising agent for the control of visceral fat accumulation (Figure 2).

## 3. Lactoferrin and Metal Metabolism

Oligo elements, including the transition metals chromium (Cr), manganese (Mn), cobalt (Co), molybdenum (Mo), copper (Cu), zinc (Zn), and iron (Fe) as well as the non-metals selenium (Se), fluorine (F), and iodine (I), play vital roles in humans and animals. Transition metals act as crucial cofactors in a variety of physiological processes, encompassing DNA replication and transcription, mitochondrial respiration, and responses to oxidative stress. Among the trace metals, Zn is the most common cofactor, followed by Fe and Mn, with approximately 60% of known enzymes relying on at least one metal cofactor. Zinc acts as an enzyme cofactor responsible for DNA synthesis and repair as well as the regulation of gene expression. Copper is involved in DNA methylation, which influences gene activity, and it is indispensable for the protein biosynthesis implicated in forming connective tissues and synthesizing neurotransmitters. Iron has a significant role in two key functions. The first one involves its role in heme, which is crucial for oxygen transport and delivery. The second function is its involvement in iron–sulfur clusters, which are essential for carrying out electron-transfer reactions. Additionally, transition metals also play a role in cellular signaling pathways, antimicrobial pathways, therapeutic treatment, and immune cell function [51,122,123,124]. Trace elements are classified as micronutrients because the body requires them in small amounts, usually less than 100 mg/day [125]. Deficient or excessive intake of essential trace elements can have negative health effects; Zn deficiency is associated with gastrointestinal disorders, Cu deficiency can cause Menkes disease, and Fe deficiency can lead to anemia. Both copper and iron imbalances have been linked to disorders such as anemia, hemochromatosis, Alzheimer’s disease, and Parkinson’s disease. Chronic exposure to high levels of trace metals can result in toxicities affecting vital organs. Disruption of redox-active metal homeostasis can lead to the formation of harmful free radicals, damaging cellular components. Even redox-inert elements like cadmium and arsenic, which have no known biological functions, can be toxic and contribute to oxidative damage and disease development by interacting with DNA and proteins [126]. For this reason, organisms have evolved mechanisms to tightly regulate transition-metal levels, ensuring a balance between uptake, chelation, distribution, and storage [6].

The primary source of trace metal elements is dietary intake, but these need to be released from food and assimilated in the gastrointestinal tract to ensure an adequate supply of micronutrients. The gut microbiota plays a significant role in the intricate relationship between the host and trace metals. While some gut bacteria and pathogens compete with the host for essential metals, the host has developed mechanisms to sequester them and to prevent bacterial access. Ongoing research aims to unravel the mechanisms underlying this competition for micronutrients. Studies have shown that certain infectious diseases can impact the plasma levels of trace elements. For instance, chronic hepatitis B infection and tuberculosis are associated with lower levels of Mn, Se, Zc, and Fe, while Cu levels are higher [127]. Cellular iron overload has been linked to disease progression and the development of infections like HIV-1 and Coronavirus Acute Respiratory Syndrome-2 (SARS-CoV-2) [128,129] since iron availability is crucial for the survival and replication of viruses and intracellular infectious agents. However, various factors such as host genetics, lifestyle choices, nutritional deficiencies, treatment interventions, and exposure to pathogens can significantly impact the diversity of the gut microbiota and disrupt the balance in the intestine. Some processed foods, especially those for infants and children, are fortified with essential trace elements. According to the recommended guidelines for infant milk powder, the added amounts of Zn, Fe, and Cu per 100 kJ are 0.12–0.36 mg, 0.07–0.3 mg, and 8.4–25 μg, respectively [130].

In this scenario, there is a chance for milk-derived bLf to come in contact with, and bind to, these metals, possibly resulting in the alteration of its physicochemical properties and functions [131].

The number of other cations in Lf molecules is relatively small compared with iron; human milk Lf contains as much as 2000 times more Fe than Mn. In human milk, about 71% of the Mn is found in the whey fraction, most of which is bound to Lf [132]. Mn/Zn-supplemented bLf has enhanced bioactivity [133,134], and Zn can be found associated with Lf in vitro under very specific conditions [135]. The Mn-Lf complex has been found to have prebiotic properties toward *Lactobacillus* strains, making it beneficial for the neonatal intestinal niche. The release of these ions has the potential to enhance the growth of *Lactobacillus* species in the intestinal lumen [136]. Notably, bLf supplemented with Mn and Zn exhibits enhanced bioactivities, as demonstrated in studies on *Legionella pneumophila* and poliovirus type 1 [133,134].

An important unsolved question concerns the potential interaction between the bLf found in infant formula and chemically supplemented Cu ions. Limited research has been conducted to determine whether Cu-supplemented Lf exhibits altered bioactivity in various cells, including immune cells. In particular, the effects of Cu-supplemented bLf on immune cells, specifically murine splenocytes and macrophages, showed that Lf with a low Cu content had different effects compared with Lf alone, including a weaker suppression of splenocyte proliferation, enhanced macrophage stimulation, increased CD4^+^/CD8^+^ ratio in T lymphocytes, and increased secretion of certain cytokines [131]. In contrast, Lf with a high Cu content exhibited opposite effects. These findings suggest that Cu supplementation can modulate Lf bioactivity, with the Cu content playing a crucial role in determining the observed changes in immune cell activity. Further research is needed to understand the underlying mechanisms of these effects [131].

While the exact Lf function in copper metabolism remains to be fully understood, it is important to acknowledge its potential interaction with Ceruloplasmin (Cp), which is the primary Cu contributor in plasma [137]. Cp plays a crucial role in iron mobilization, converting ferrous iron to ferric iron, which is then bound by Tf. Various techniques, such as small-angle X-ray scattering and fluorescence measurements, have been used to study Cp’s interactions with Tf and Lf [138], shedding light on the long-studied connection between copper and iron. The formation of complexes between Cp and these proteins, facilitating the direct transfer of ferric iron within the complex, has been observed. Moreover, investigations using plasmon resonance and Hummel–Dreyer chromatography techniques demonstrate a notable distinction in the interactions between Cp and Lf/Tf. A strong interaction has been observed between Cp and apo-Lf, with a dissociation constant (K_d_) of 7.4 μM, whereas a weaker interaction between Cp and apo-Tf has been found (K_d_~51.3 μM) [138]. These results suggest that under physiological conditions, the Cp-Lf interaction may remain stable, while the Cp-Tf interaction is less likely to occur. The presence of the Cp-Lf complex in the plasma and exudates of individuals with inflammatory diseases underpins Lf’s distinctive role in iron metabolism, suggesting a role of the complex in oxidizing the ferrous iron released during inflammatory reactions [138].

Regarding the protective role of Lf toward heavy-metal-mediated toxicity, combined findings suggest that Lf could represent a promising therapeutic approach for male infertility treatment due to its ability to reduce the uptake of toxic metals like Cd. Lf supplementation alleviated spermatogenetic dysfunction induced by Cd exposure in mice, regulating apoptosis, autophagy, and oxidative-stress-related factors [139]. In Caco-2 cells, Lf reduced Cd uptake by binding to the cell membrane and reducing the cell-surface charge [140]. In a separate pilot study, a negative correlation was observed between urinary arsenic levels and Lf levels in breast milk. Exposure to arsenic had an impact on breast milk composition, likely due to the secondary effects of arsenic-mediated estrogen functions [141].

Overall, the interactions of Lf with transition metals indicate its involvement in maintaining metal balance and its potential as a carrier for these metals. However, more research is needed to fully understand the mechanisms and implications of such interactions in relation to human health and infectious diseases.

### Lactoferrin and Iron Metabolism

A separate section needs to be devoted to iron homeostasis, as this is the most characterized metal pathway indirectly and directly regulated by Lf.

Preserving an optimal level of iron in the body is crucial for supporting various physiological processes in humans. The body contains approximately 3–4 g of iron, with the majority found in hemoglobin as heme. A daily intake of around 20 milligrams of iron is necessary to support the production of new red blood cells. Iron consumption from dietary sources is vital throughout life, starting from fetal development to adulthood. However, only a small portion of dietary iron, about 10%, is absorbed due to its limited solubility. Breast milk contains Lf, which is a significant iron source during early life. However, not all Lf in human milk is bound to iron. Reconstituted dried cow’s milk, commonly used as an alternative source, also contains the glycoprotein but in lower concentrations compared with human milk [142].

After the weaning period, the main sources of bioavailable iron include heme iron from hemoglobin and myoglobin, as well as non-heme iron from plant and animal foods. The mechanisms involved in heme iron absorption are not yet fully understood, although one pathway involves heme carrier protein 1 (HCP1) [143]. Non-heme iron is absorbed by enterocytes through a vesicular transport system, where it initially binds to the brush-border membrane and is subsequently released into the cytoplasm by divalent metal transporter 1 (DMT1) [144]. Once inside the cell, ferrous iron is utilized for cellular functions or bound by chaperone proteins, while excess iron is stored as ferritin (Ftn) or hemosiderins. For iron to be biologically active, it must be released from Ftn, usually through lysosomal degradation [145]. Ferrous iron can also be exported from the cell into the bloodstream through ferroportin (Fpn), in conjunction with a multicopper ferroxidase, which varies depending on the cell type. Hephestin (Heph) is found exclusively in enterocytes, in the CNS [146], and in the kidney [147]. Cp, introduced in the previous section, is an acute-phase plasma protein with various functions, including copper transport, oxidation of biological amines, and antioxidant activity [148]. Alternative splicing gives rise to a membrane-anchored form of Cp, which is mainly expressed in astrocytes, hepatocytes, macrophages, and retinal epithelial cells [149,150]. This form of Cp plays a crucial role in facilitating iron release, as described previously. Iron is then transported to target cells by binding to Tf and utilizing the Tf receptor 1 (TfR1) complex.

The body preserves iron balance through a combination of cellular and systemic mechanisms, working together to regulate the absorption, utilization, storage, and release of iron to maintain appropriate levels. The hormone hepcidin plays a crucial role in regulating iron metabolism at a systemic level. By reducing the absorption and release of iron from cells, hepcidin decreases the amount of iron in the bloodstream [151]. The production of this hormone is influenced by factors such as overall iron levels, red blood cell production, low oxygen levels, and inflammation [142]. Inflammatory signals trigger the sequestration of iron, making it less available to other tissues. Conversely, conditions characterized by iron deficiency or increased red blood cell production require the suppression of hepcidin production. The regulation of iron balance involves complex interactions to ensure an adequate supply of iron while preventing both excess and deficiency.

Lf iron saturation plays a critical role in controlling the amount of iron within cells and regulating iron-related proteins, including those found in cancer cells [20]. Computational models further suggest that both hLf and bLf can physically interact with TfR1, indicating a potential shared mechanism for iron absorption when bound to Tf [152]. Interestingly, Lf has a unique ability to access the nucleus of cells and interact with specific and non-specific DNA sequences, functioning as a transcription factor [153,154]. It has been found to facilitate the nuclear translocation of nuclear factor erythroid-2-related factor 2 (Nrf2) [14,155], a crucial regulator of the cellular antioxidant response. Moreover, Lf contributes to the upregulation of Fpn, the sole iron exporter, thus actively contributing to reverse anemia-related sequelae [6]. Inflammatory conditions or infections can lead to the degradation of the iron exporter Fpn by pro-inflammatory cytokines, resulting in the reduced iron transportation from cells to the bloodstream [156]. This causes an overload of iron within cells and systemic iron deficiency, which can lead to anemia or anemia of inflammation (AI) [156,157]. Iron deficiency without anemia is characterized by low levels of serum iron and Ftn, while iron-deficiency anemia is associated with low red blood cell and hemoglobin levels. AI is characterized by low hematological parameters, elevated Ftn levels, and increased levels of pro-inflammatory cytokines like IL-6. Restoring a balance of iron between tissues/secretions and blood is essential in counteracting the persistence of inflammation and infections and in maintaining iron homeostasis. New therapeutic approaches for addressing AI focus on Fpn agonists or hepcidin antagonists [158,159]. In this context, bLf has been shown to be a promising candidate that is able to reduce IL-6 levels and increase Fpn expression, thus rebalancing iron levels [30,160,161,162,163]. Clinical studies have demonstrated the effectiveness of bLf supplementation in treating anemic pregnant women [30,163,164], patients with hereditary thrombophilia [30,163], and those with minor β-thalassemia [165]. Compared with traditional iron therapies, bLf administration led to significant improvements in red blood cell counts, hemoglobin levels, total serum iron, and Ftn levels, and a decrease in IL-6 and hepcidin concentrations. The timing of bLf administration can influence its effectiveness, with better outcomes observed when taken before meals to avoid degradation by proteases [34]. In contrast, traditional ferrous sulfate treatment failed to restore iron parameters and lower IL-6 and hepcidin levels [164]. Mechanistic studies have shown that bLf treatment can reverse the degradation of Fpn and suppress pro-inflammatory activity in infected enterocytes [160]. It also promotes a shift from the pro-inflammatory M1 to the anti-inflammatory M2 macrophage phenotype [166,167]. This shift encourages the production of anti-inflammatory cytokines, increases the expression of Fpn, and restores the natural export of iron from macrophages to the bloodstream [161,162]. Additionally, bLf can regulate other iron-related proteins like TfR1, Cp, and cytosolic Ftn, further contributing to iron balance in the body [162]. Animal studies have also demonstrated the protective effects of bLf in experimental hemorrhagic anemia, resulting in increased levels of hemoglobin and serum iron [166].

The connection between imbalanced iron levels and inflammation has been established as a negative factor in patients infected with hepatitis B virus (HBV), hepatitis C virus (HCV) [128], and SARS-CoV-2 [167], as well as in patients infected with HIV-1 [168]. In SARS-CoV-2 and HIV-1 patients, elevated Ftn levels are linked to a worsened prognosis [167,168,169]. Notably, elevated levels of serum Ftn can be harmful to the body, leading to hepatocellular death and increased levels of free iron. This excess iron can worsen inflammatory conditions, trigger ferroptosis, and potentially result in multiple-organ failure. In particular, Ftn levels are found to be high in severe cases of COVID-19, where the presence of free iron can contribute to hypercoagulation, further complicating the disease [129]. Excessive iron levels in individuals with HIV can lead to increased viral replication, inflammation, and unfavorable outcomes. It is not unusual for HIV patients to have high Ftn levels, which can contribute to viral replication [169]. A preliminary study was conducted to investigate the effectiveness of oral and intranasal liposomal bLf in treating COVID-19 patients with asymptomatic and mild-to-moderate symptoms. These patients experienced a quicker recovery from clinical symptoms and demonstrated reductions in serum Ftn, thus demonstrating that Lf may have potential in managing mild-to-moderate and asymptomatic cases of COVID-19 [170].

In addition, iron disorders are associated with the action of non-structural viral proteins [14,152] such as virulence factors and neurotoxins [171,172]. Two in vitro studies specifically looked at the action of Lf against iron disorders induced by viral proteins, such as the spike protein of SARS-CoV-2 and the tat protein of HIV-1.

One study showed that the spike protein of SARS-CoV-2 disrupts iron-handling proteins, and that Lf can rebalance this dysregulation and reduce inflammation. These findings suggest that Lf could be used as a therapy in the early stages of SARS-CoV-2 infection [152].

The other study demonstrated the role of bLf in reducing oxidative stress and oxidative damage in cells expressing the tat protein of HIV-1. Additionally, Lf restored the normal function of iron uptake in cells expressing the viral protein. However, the effects of the glycoprotein varied depending on its iron saturation and emphasized the need for the quality standardization of Lf products [14]. The studies conducted so far on Lf show promising insights into its therapeutic potential, particularly in the context of viral infections and iron dysregulation. However, further research is necessary to confirm its effectiveness in treating such infections in real-life settings. These findings open possibilities for future investigations and highlight the potential therapeutic benefits of Lf in infection conditions.

In conclusion, Lf holds promise as a potential solution for managing iron levels and for protecting against iron-related disorders, and as a versatile therapeutic agent to address the interconnected relationship between infection, inflammation, and iron disorders.

## 4. Conclusions

Lactoferrin has being acknowledged for its multitargeting and multifaceted activities, high tolerability, and ease of production, which have made it of great interest to the nutraceutical industry.

To date, several preparations of Lf, mainly from the bovine source, are commercially available, such as in infant formulae for newborns and in fortified beverages and foods for adults. The consumption of these products is safe, and different benefits for human health, including the promotion of oral and gastrointestinal homeostasis [173,174,175] the control of iron absorption and balance [6,176], as well as the regulation of glucose and lipid metabolism, have been reported.

Of note, several clinical studies have highlighted the actual efficacy of such Lf preparations in some inflammatory [30,177,178] and infectious conditions [179,180,181], including, most recently, COVID-19 [170,182,183].

As evidenced by this review, Lf can exert a dual role in human metabolism by both boosting it under physiological conditions or resetting it under pathological ones. Different mechanisms have been demonstrated to be directly or indirectly involved in such action. The ability of Lf to bind glucose could partially explain the role of this glycoprotein in influencing its uptake by intestinal receptors or its systemic bioavailability. Moreover, it has been shown that whey proteins stimulated the translocation of glucose transporter 4 (GLUT4) to the plasma membrane in muscle tissue independently of insulin secretion [184], and that Lf itself can reverse the GLUT4 downregulation triggered by HFD [94]. This suggests another potential mode of hypoglycemic action by Lf; however, the detailed mechanism remains to be determined in future studies.

Lf can interact with or compete for cell receptors naturally triggered by different ligands. For instance, Lf was demonstrated to inhibit cholesterol accumulation in macrophages by interfering with the binding of acetylated or oxidized LDL to scavenger receptors [185]. Moreover, intravenous Lf injection inhibited the LRP-mediated uptake of ApoE-containing chylomicron remnants and VLDL [186,187].

Finally, Lf has been widely acknowledged for its ability to directly or indirectly influence metal metabolism, including of iron, one of the main participants in energy production and which is responsible for both physiological and pathological interrelated processes.

As suggested from the data presented, the mechanism of Lf action in reprogramming human metabolism involves several processes, including the regulation of glucose and lipid absorption, improvement of insulin production and signaling, inhibition of adipogenesis, elevation of HDL cholesterol along with the decrease in oxidized LDL cholesterol forms, reduction in inflammation, and oxidative stress related to metabolic syndrome.

Overall, even though more studies in humans are required, Lf can be recommended as a promising, completely non-toxic natural adjuvant that can be applied in the long-term prophylaxis and therapy of metabolic disorders such as insulin resistance/type II diabetes and metabolic syndrome.

## Figures and Tables

**Figure 1 ijms-24-15925-f001:**
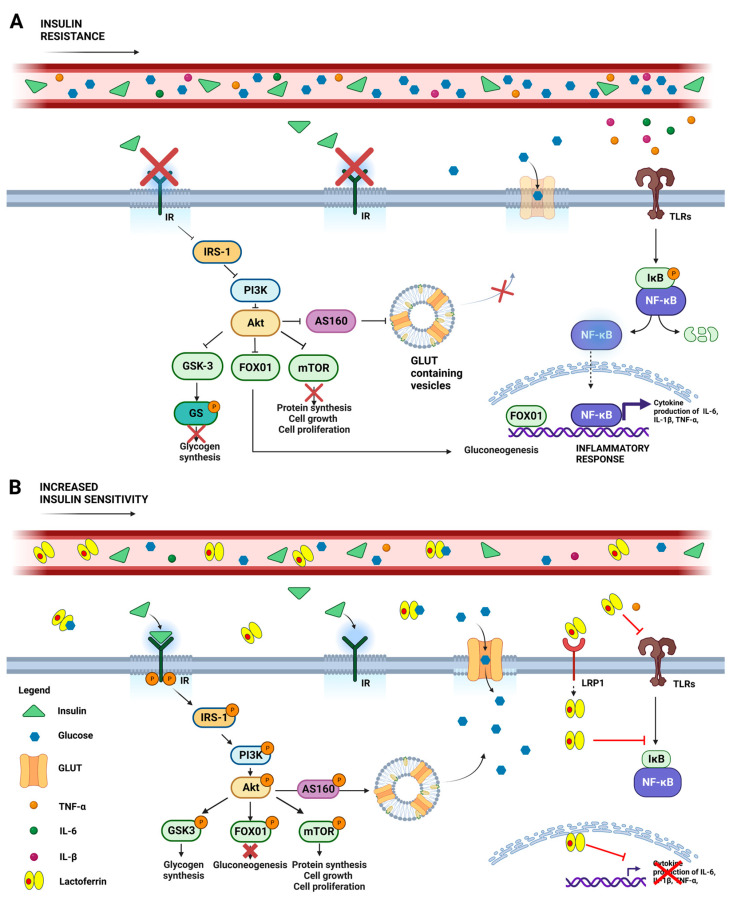
Schematic representation of systemic glycemic disorders in the absence (**A**) or presence (**B**) of lactoferrin. (**A**) Insulin resistance is a clinical condition where insulin’s ability to promote glucose absorption and utilization is impaired, resulting in elevated blood glucose levels. This condition hampers IR autophosphorylation, which impairs the IRS-1/PI3-kinase/AKT pathway and results in aberrant downstream signals, including GSK3 activation and the consequent inhibition of glycogen synthesis via the phosphorylation of GS; the reduction of mTOR-mediated protein synthesis, cell growth and proliferation; as well as FOX01nuclear translocation, which promotes gluconeogenesis and the inflammatory response. Stimulation of TLRs by TNF-α, IL-6, and IL-1β exacerbates insulin resistance through the inflammatory IKβ/NF-κB pathway, increasing cytokine expression. (**B**) Lactoferrin treatment counteracts these detrimental effects by boosting insulin binding to IR, thereby activating the IRS-1/PI3-kinase/Akt pathway. Akt activation results in AS160 phosphorylation, prompting GLUT to relocate from intracellular vesicles to the cell membrane, thus improving glucose uptake. Simultaneously, Akt-mediated FOX01 phosphorylation inactivates its nuclear translocation, while GSK3 and mTOR phosphorylation promote glycogen and protein synthesis, respectively. In addition, the protective effect of Lf could be explored by its ability to bind glucose and by virtue of its anti-inflammatory activity, thus counteracting TLR-mediated detrimental signaling. Abbreviations: insulin receptor (IR); insulin receptor substrate-1 (IRS-1); phosphatidyl inositol 3-kinase (PI3-Kinase); Akt, also known as protein kinase B; GSK3 (glycogen synthase kinase-3); GS (glycogen synthase); FOX01 (forkhead-box protein 01); Toll-like receptors (TLRs); tumor necrosis factor-α (TNF-α), interleukin-6 (IL-6); interleukin-1β (IL-1β); nuclear factor kappa B (NF-κB); inhibitor of NF-κB (IKβ); glucose transporter (GLUT). Created with BioRender.com (accessed on 15 June 2023).

**Figure 2 ijms-24-15925-f002:**
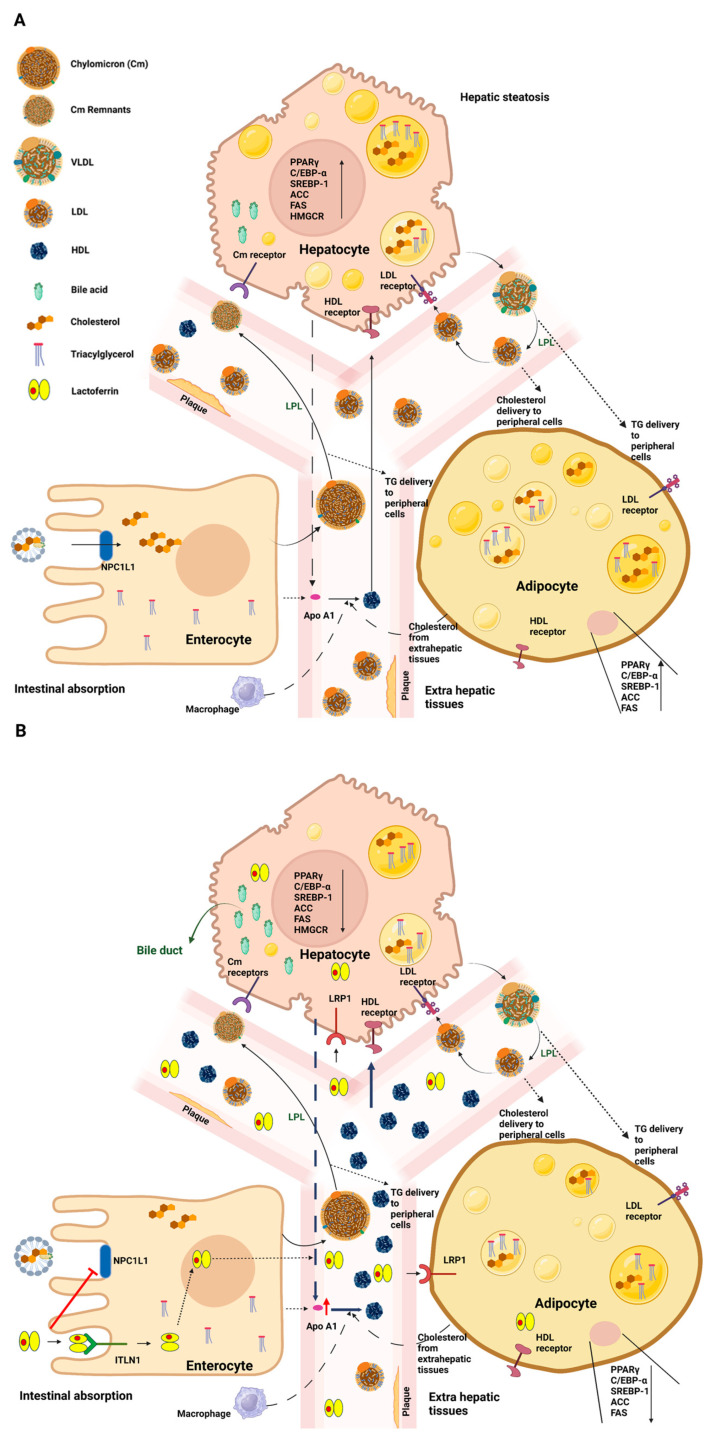
Schematic representation of systemic lipidic disorders in the absence (**A**) or presence (**B**) of lactoferrin. (**A**) Dietary fats, after emulsification and hydrolysis by bile acids, are taken up by enterocytes. Luminal cholesterol is transported across the brush border via NPC1L1 and, together with other lipids, is packed into nascent Cm particles. Once in the bloodstream, the Cm particles undergo transformations and are subject to the influence of LPL, which assists in converting Cm triglycerides into fatty acids and glycerol. Subsequently, hepatocytes internalize the Cm remnants through Cm receptors. In the context of systemic lipid disorders, upregulation of PPARγ, C/EBP-α, and SREBP-1 pathways and of other adipogenesis markers, such as ACC, FAS, and HMGCR, occurs, along with an increase in the levels of intracellular lipid droplets. Hepatic cells release abnormal quantities of VLDL, which are converted to LDL by LPL to release triglycerides to various tissues. This process elevates circulating LDL levels, contributing to the development of atherosclerotic plaques. On the other hand, low levels of circulating HDL are recorded. (**B**) Upon oral Lf treatment, a reduction in cholesterol absorption in the intestine is observed. In accordance with the downregulation of C/EBP-α, PPARγ, and SREBP-1pathways and the reduction of specific lipogenic enzymes, including FAS and ACC, a decrease in lipid droplet levels and circulating LDL is found. Interestingly, higher serum levels of HDL are observed. See text for further details. Abbreviations: Neiman–Pick C1-like 1 (NPC1L1); chylomicron (Cm); lipoprotein lipase (LPL); peroxisome proliferator-activated receptor γ (PPARγ); CCAAT/enhancer-binding protein (C/EBP-α); sterol-regulatory-element-binding protein-1 (SREBP-1); acetyl-CoA carboxylase (ACC); fatty-acid synthase (FAS); 3-hydroxy-3-methyl glutaryl CoA reductase (HMGCR), very-low-density lipoprotein (VLDL); low-density lipoprotein (LDL); high-density lipoprotein (HDL). Created with BioRender.com (accessed on 15 June 2023).

## Data Availability

No new data were created. All data supporting reported results can be found in analyzed publications.

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
