# Peer review of "To Boost or to Reset: The Role of Lactoferrin in Energy Metabolism"

_ijms, 2023, doi:10.3390/ijms242115925_

Round 1
Reviewer 1 Report
Comments and Suggestions for Authors
The aim of Ianiro et al.'s manuscript is to provide a comprehensive review of lactoferrin protein, covering its biochemical, molecular, and physiological aspects and its role as a nutritional supplement in regulating human physiology and metabolism. However, the manuscript lacks a logically organized analysis of lactoferrin functions, and the cited evidence is limited without expert analysis. Moreover, the review includes lengthy presentations of general biological concepts that are not directly relevant and take up significant space (e.g. lines 202-250, lines 314-363). As a result, the current form of the manuscript is not suitable for publication.
Author Response
The aim of Ianiro et al.'s manuscript is to provide a comprehensive review of lactoferrin protein, covering its biochemical, molecular, and physiological aspects and its role as a nutritional supplement in regulating human physiology and metabolism. However, the manuscript lacks a logically organized analysis of lactoferrin functions, and the cited evidence is limited without expert analysis. Moreover, the review includes lengthy presentations of general biological concepts that are not directly relevant and take up significant space (e.g. lines 202-250, lines 314-363). As a result, the current form of the manuscript is not suitable for publication.
We thank the Reviewer for his/her comments. The aim of the manuscript was not to provide an exhaustive analysis of Lf functions, there are many excellent reviews already available. We did our best to eliminate the overly didactic parts, and the lines indicated by the Reviewer were essentially omitted in the revised text.
Reviewer 2 Report
Comments and Suggestions for Authors
Dear authors,
I would like to recommend shorten phrases in some cases, as e.g. lines 350 - 362.
I recommend also to include in Conclusions the real possibility Lf to be used as a therapy of inflammation, infections (especislly virsl diseases as Covid-19, etc.). There are data about successes in medicsl practice - treating and preventing with Lf viral diseases (e.g. products of Doppel Hertz).
Best regards!
Author Response
Dear authors,
I would like to recommend shorten phrases in some cases, as e.g. lines 350 - 362.
I recommend also to include in Conclusions the real possibility Lf to be used as a therapy of inflammation, infections (especially viral diseases as Covid-19, etc.). There are data about successes in medical practice - treating and preventing with Lf viral diseases (e.g. products of Doppel Hertz).
We thank the Reviewer for the helpful comments. The manuscript has been amended accordingly.
Round 2
Reviewer 1 Report
Comments and Suggestions for Authors
The revised manuscript is satisfactory and meets the standards for publication.